# Influence of the Duration and Timing of Data Collection on Accelerometer-Measured Physical Activity, Sedentary Time and Associated Insulin Resistance

**DOI:** 10.3390/ijerph18094950

**Published:** 2021-05-06

**Authors:** Tanja Sjöros, Henri Vähä-Ypyä, Saara Laine, Taru Garthwaite, Eliisa Löyttyniemi, Harri Sievänen, Kari K. Kalliokoski, Juhani Knuuti, Tommi Vasankari, Ilkka H. A. Heinonen

**Affiliations:** 1Turku PET Centre, University of Turku and Turku University Hospital, 20521 Turku, Finland; saalaiy@utu.fi (S.L.); taru.garthwaite@utu.fi (T.G.); kari.kalliokoski@tyks.fi (K.K.K.); juhani.knuuti@tyks.fi (J.K.); ilkka.heinonen@utu.fi (I.H.A.H.); 2The UKK-Institute for Health Promotion Research, Kaupinpuistonkatu 1, 33500 Tampere, Finland; henri.vaha-ypya@ukkinstituutti.fi (H.V.-Y.); harri.sievanen@ukkinstituutti.fi (H.S.); tommi.vasankari@ukkinstituutti.fi (T.V.); 3Department of Biostatistics, University of Turku, 20014 Turku, Finland; eliisa.loyttyniemi@utu.fi; 4Faculty of Medicine and Health Technology, Tampere University, 33720 Tampere, Finland; 5Rydberg Laboratory of Applied Sciences, University of Halmstad, 30118 Halmstad, Sweden

**Keywords:** sedentary behavior, insulin sensitivity, accelerometry, measurement accuracy, measurement error, data variability

## Abstract

Accelerometry is a commonly used method to determine physical activity in clinical studies, but the duration and timing of measurement have seldom been addressed. We aimed to evaluate possible changes in the measured outcomes and associations with insulin resistance during four weeks of accelerometry data collection. This study included 143 participants (median age of 59 (IQR9) years; mean BMI of 30.7 (SD4) kg/m^2^; 41 men). Sedentary and standing time, breaks in sedentary time, and different intensities of physical activity were measured with hip-worn accelerometers. Differences in the accelerometer-based results between weeks 1, 2, 3 and 4 were analyzed by mixed models, differences during winter and summer by two-way ANOVA, and the associations between insulin resistance and cumulative means of accelerometer results during weeks 1 to 4 by linear models. Mean accelerometry duration was 24 (SD3) days. Sedentary time decreased after three weeks of measurement. More physical activity was measured during summer compared to winter. The associations between insulin resistance and sedentary behavior and light physical activity were non-significant after the first week of measurement, but the associations turned significant in two to three weeks. If the purpose of data collection is to reveal associations between accelerometer-measured outcomes and tenuous health outcomes, such as insulin sensitivity, data collection for at least three weeks may be needed.

## 1. Introduction

Sedentary behavior (SB) and lack of moderate-to-vigorous physical activity (MVPA) have been identified as important predictors of metabolic disorders, type 2 diabetes, cardiovascular diseases, and premature mortality [1,2,3,4,5,6]. Originally, these associations were discovered using subjective self-assessment tools, but during the last decade more objective methods, such as accelerometers or other devices, have become more accessible and therefore more commonly used methods to determine physical activity (PA) or SB in quantitative clinical studies examining the associations between SB and health outcomes [7]. The optimal way to measure PA and SB with accelerometers has been discussed in past papers [8,9,10,11,12]. As a result, guidelines for appropriate device placement, data analysis approaches, and cut-points have been proposed [11]. However, the duration of data collection has remained a rarely addressed question [13]. At present, there is very little published evidence on how long data collection should take place in order that it may be considered to represent ‘habitual’ PA or SB within specific study samples.

The most commonly used duration of accelerometer data collection in a clinical study is 7 days. Little discussion has emerged about whether or not this is a sufficient duration to measure individuals’ typical SB and PA behaviors [13]. In earlier trials, the durability of the batteries and storage capacity of the devices were limiting factors, and thus one week was established as the typical length of the measurement most likely due to practical reasons. In large, population-based cohorts, sufficient accuracy can be achieved with relatively shorter measurement periods, but it is questionable whether one week of accelerometry is long enough a period to capture long-time average behaviors in smaller cohorts or actual changes of behavior in intervention studies. This may be a weakness in the current evidence on SB and health, as it is also possible that wearing a measurement device actually has an impact on one’s SB and PA behaviors. There is the possibility of both random and systematic errors in the accelerometry if the measurement period is too short.

The question concerning random error and regression dilution bias has been addressed by the intraclass correlation coefficient (ICC) or mean absolute percentage error of measurement (MAPE). There appears to occur considerable intraindividual variation in daily PA and SB. Aadland & Ylvisåker reported that at least 15 consecutive days are needed to reliably estimate sedentary time in adults (ICC ≥ 0.8) compared to 21 days of measuring with accelerometers worn on the hip [14]. In older adults, 5 days of SB monitoring was needed to reach ICC ≥ 0.8, and 11 days to reach ICC ≥ 0.9 compared to 21 days of monitoring [15]. However, it remains unresolved whether 21 days is a sufficient duration to represent habitual everyday SB or PA over a longer period, e.g., a year. Measured by a pedometer, at least 30 consecutive days of measurement were needed to reach MAPE lower than 10% or at least 5 consecutive days to reach ICC ≥ 0.8, and 14 days to reach ICC ≥ 0.9, compared to a full 365 days of monitoring [16]. Nevertheless, calculating proportions of different behaviors may reduce the number of data collection days needed to achieve sufficient reliability in cross-sectional studies [14].

This study aimed at investigating systematic measurement effects on measured PA and SB outcomes and related health associations during four weeks of data collection by accelerometry. The participants were instructed to maintain their habitual physical activity and sedentary behaviors during the measurement. However, it is possible that being subjected to a measurement does have an impact on one’s behavior, even if unintended. The awareness of being monitored can have an intervention effect on the participants’ behavior that can be compared to the placebo effect [17,18]. We hypothesized that the possible intervention effect of wearing the measurement device would weaken during a prolonged data collection period, thus leading to systematic differences in the measured SB and PA between weeks 1, 2, 3 and 4.

As originally described by Matthews et al., the homeostasis model assessment for insulin resistance (HOMA-IR) can be used as a surrogate measure of insulin resistance [19]. In our previous study, we observed a difference in the associations between HOMA-IR and measured SB and PA when analyzing data from the full four weeks of measurement vs. only the first week [20]. Consequently, we wanted to investigate how many weeks of data collection was needed to reveal this association.

Therefore, in this study, we aimed at measuring several parameters of SB and PA, including standing, for four consecutive weeks and comparing the average results of weeks 1, 2, 3 and 4 with each other. Additionally, we wanted to investigate whether data collection during different seasons had an impact on the average results. Moreover, we evaluated the averages both as units of time and as proportions of wear time. SB and PA behaviors were measured with hip-worn tri-axial accelerometers, in 6 s (0.1 min) epochs over the whole four-week period, to obtain the best possible precision in describing individual SB and PA levels. Furthermore, we investigated the associations between HOMA-IR and cumulative means of different SB and PA behaviors during weeks 1 to 4.

## 2. Materials and Methods

This study was a one-arm explorative observational study consisting of the screening phase of an intervention study registered at ClinicalTrials.gov (NCT03101228). The study participant recruitment and data collection were conducted at the Turku PET Centre, Turku, Finland between April 2017 and May 2019. This study was conducted according to good clinical practice and the Declaration of Helsinki. Participants gave their informed consent before entering the study. The study was approved by the Ethics Committee of the Hospital District of Southwest Finland (16/1810/2017).

The participants in this study were recruited from the local community by newspaper advertisements and bulletin leaflets as previously reported [20]. The inclusion criteria for selecting the participants were: Age 40–65 years, body mass index (BMI) 25–40 kg/m^2^, and, according to self-reporting, the participants should not meet the current recommendations for physical activity and should sit for a major portion of the day. The exclusion criteria were: history of a cardiac event, diagnosed diabetes, abundant use of alcohol (according to national guidelines), use of narcotics, smoking of tobacco or consuming of snuff tobacco, inability to understand written Finnish, and any chronic disease or condition that could create a hazard to the participant’s safety or endanger the study procedures.

The eligible volunteers were interviewed, and during the interview they received an accelerometer, which they were instructed to wear on the right hip for four consecutive weeks, starting the following morning. They were instructed to wear the accelerometer during all waking hours, except for activities where the device would be exposed to water. Moreover, they were advised to maintain their habitual activities and ways of life during the measurement. To ensure continuous measurement, after two weeks the participants visited the research center again to receive a new device with fresh batteries, after which the measurement was continued as before for two more weeks. All the participants with valid accelerometer data from at least two weeks were included in this analysis.

SB and PA were measured for four weeks with hip-worn tri-axial accelerometers (UKK AM30, UKK-Institute, Tampere, Finland) using a digital triaxial acceleration sensor (ADXL345; Analog Devices, Norwood, MA, USA). The small and light (37 × 27 × 9 mm^3^, 9.3 g) device is attached to a flexible belt and allows free movement. The collected accelerometer data was analyzed in six-second epochs, using a validated mean amplitude deviation (MAD) algorithm [21]. The epoch-wise MAD values were converted to metabolic equivalents (METs) (3.5 mL/kg/min of oxygen consumption) [21]. Light physical activity (LPA) was defined as a MET value higher than or equal to 1.5 and less than 3.0 (MAD value between 22.5–91.5 milligravity units (mg)), MVPA as a MET value higher than or equal to 3.0 (MAD over 91.5 mg). The body posture (i.e., lying, sitting, standing) was determined with angle for posture estimation (APE) algorithm only for the epochs with MAD value lower than 22.5 mg [22]. Breaks in sedentary time represent the number of SB periods during which the one-minute exponential moving average of the estimated MET value was less than 1.5, and which was finished by a clear vertical acceleration and subsequent standing position or movement [22]. Wear time of 10–19 h/day and a minimum of 3 days of measurement/week were considered valid. The mean duration in different categories of PA and SB was calculated individually for weeks 1, 2, 3 and 4. Additionally, mean proportions of SB and different PA categories per day were calculated, and presented as percentage of wear time. Moreover, the cumulative means of different categories of PA and SB during weeks 1, 2, 3 and 4 were calculated. All the available data with 10–19 h/day wear time were used in calculating the cumulative means, including the weeks with less than 3 valid days, provided that there were at least two consecutive weeks with at least 3 valid days.

Fasting blood samples were drawn at the participants’ most convenient time during the accelerometer data collection period. Plasma insulin was determined by electrochemiluminescence immunoassay (Cobas 8000 e801, Roche Diagnostics GmbH, Mannheim, Germany) and glucose by an enzymatic reference method with hexokinase GLUC3 (Cobas 8000 c702, Roche Diagnostics GmbH, Mannheim, Germany) at the Turku University Hospital Laboratory. HOMA-IR index was calculated with the formula glucose x insulin/22.5. 

The differences in the accelerometer results during weeks 1, 2, 3 and 4 were analyzed by mixed model for repeated measures with two categorical variables: time as a within-subject factor and sex as a between-subject factor, and the interaction term (time*sex). Sex was included in the model in order to examine if men behave differently from women during the data collection period. The differences between accelerometer results measured during winter (November–March) and summer (April–October) were analyzed by two-way ANOVA. The associations between HOMA-IR and different durations of accelerometer measurements were tested by linear models with one categorical variable (sex) and three continuous variables (age, BMI, and accelerometer outcome) in the model, and therefore the association between continuous factors and the response is described as a slope. Logarithmic (log10) transformations were performed when necessary to achieve normal distribution of the data. The normal distributions of the residuals were examined visually, and sensitivity analyses were performed when needed via the leave-one-out method to assure the robustness of the findings. If not otherwise stated, data are expressed as means (SD) or means with 95% confidence interval, when applicable. In the case of a skewed distribution, median (IQR) is presented. The level of statistical significance was set at 5%. All analyses were carried out with SAS 9.4 and JMP pro 13.1 for Windows (SAS Institute Inc., Cary, NC, USA).

## 3. Results

In total, 263 participants volunteered, of whom 102 women and 41 men were found to be eligible and completed the accelerometer measurements with at least two valid weeks of data collection. The basic characteristics of the study subjects are presented in Table 1. The mean accelerometer wear time was 14.38 (SD 1.04) h/day, and the mean duration of the data collection was 25 (SD 3) days. The duration varied from 10 to 28 days and 91% of the participants had valid data collected during all four weeks. The participants spent 66.8 (SD 8.1) % of the total accelerometer wear time engaged in sedentary activities.

### 3.1. Changes in Measured SB and PA over Four Weeks

Measured sedentary time decreased during the measurement period (Figure 1). Measured mean sedentary time was 9.88, 9.82, 9.73, and 9.57 h during weeks 1–4, respectively. The fourth week differed significantly from weeks 1 and 2 (*p* = 0.0033 and 0.021, respectively). Men had significantly more sedentary time compared to women, with the 4-week average being 10.09 h/day for men and 9.41 h/day for women. 

In LPA there was a significant sex*time interaction (Figure 1). During week 3, men had significantly less LPA compared to women (*p* = 0.018). In men, mean daily LPA during week 3 (1.59 h) was significantly less than during weeks 1 (1.73 h, *p* = 0.023) and 4 (1.76 h, *p* = 0.0066). In women, mean daily LPA during week 4 (1.80 h) was significantly lower than during week 1 (1.88 h, *p* = 0.045).

No significant changes occurred in standing time, MVPA, or breaks in sedentary time during measurement weeks 1–4 (Figure 1). Accelerometer wear time decreased after the first week, with no overall difference between sexes or time*sex interaction (Figure 1). The average wear time during weeks 1–4 was 14.59, 14.34, 14.26, and 14.19 h/day, respectively (*p* = 0.0090, 0.0004, and <0.0001 for week 1 vs weeks 2, 3 and 4, respectively).

### 3.2. Proportions of Different Activity Categories

SB, standing, or MVPA proportions did not change during the four weeks of measurement (Figure 2). However, there was a near significant (*p* = 0.087) difference in SB proportion between weeks 2 and 4. There was a significant sex*time interaction in LPA proportion (Figure 2). During week 3, men had a significantly lower LPA proportion (11.3%) than women (12.8%, *p* = 0.035). In men, LPA proportion during week 4 (12.6%), was significantly higher than during week 3 (*p* = 0.023). In women, there were no differences in LPA proportions during weeks 1–4 (Figure 2). 

### 3.3. Seasonal Variation

There were some differences in the four-week averages of SB and PA measured during winter (n = 69) and summer (n = 74) (Figure 3 and Figure 4). More breaks in sedentary time were measured during summer (mean 30 breaks/day) compared to winter (mean 27 breaks/day) (Figure 3). Women had significantly more standing time during summer compared to winter (*p* < 0.0001), and also, compared to men, both during winter and summer (*p* = 0.014 and <0.0001, respectively). Both men and women had more MVPA during summer compared to winter (Figure 3). 

The proportion of measured sedentary time was significantly lower during summer months than during winter months (Figure 4). As was the case with PA time, women had a greater proportion of standing time during summer (*p* = 0.0002), and both sexes had a greater proportion of MVPA during summer compared to winter (Figure 4).

### 3.4. Duration of Accelerometer Data Collection in Predicting Insulin Resistance

Based on accelerometer data collected during the first measurement week, no association was found between HOMA-IR and SB proportion or LPA, whereas the association between HOMA-IR and MVPA was significant (Table 2 and Table 3). The association between HOMA-IR and SB proportion turned significant in two weeks and LPA time in three weeks of accelerometry, i.e., the associations with cumulative means of two and three weeks of accelerometry were significant (Table 2 and Table 3). The associations between HOMA-IR and sedentary and standing time, as well as standing and LPA proportions, remained non-significant throughout the four weeks of accelerometry. However, in sensitivity analyses where one participant with extreme outlier values in the residuals was excluded, the LPA proportion turned significant within two weeks (*p* = 0.091, 0.049, 0.028, and 0.025 in 1, 2, 3 and 4 weeks, respectively). Sex and BMI were significant in all of the models with BMI as the strongest predictor of HOMA-IR. Additionally, the models were tested with the season included as a categorical variable, but it was not significant in any of the models and it did not essentially change the interpretation of the results (data not shown). Therefore, the season was not included in the final linear models.

## 4. Discussion

In this study, we demonstrated that accelerometer-measured sedentary time is related to the timing and duration of the measurement, and therefore the data collection period should be sufficiently long. Additionally, the measured LPA varied during prolonged data collection, whereas measured MVPA seemed more robust and less variable, and associations to insulin sensitivity could be detected with a relatively short measurement period. However, there was seasonal variation in MVPA, which should also be carefully considered in planning accelerometer data collection.

In the present study, we found a significant decrease in accelerometer-measured sedentary time after 3 weeks of measurement. However, there was also a significant decrease in accelerometer wear time after the first week, which may, at least in part, explain the change in measured SB, even if the changes in SB and wear time were not completely linear. Calculating proportions of SB and different PA categories diluted this time-effect and we did not observe any differences in measured SB proportions between weeks 1, 2, 3 and 4, which is in line with previous arguments [14]. However, there was a trend towards a difference in SB proportion between weeks 2 (68.4%) and 4 (67.4%). This may indicate that with a bigger sample size we could also have observed a difference in sedentary proportions during different weeks, but the clinical significance of a 1% difference remains disputable.

There occurs considerable intraindividual variation in daily PA and SB. The possibility for random error has been addressed by the “Spearman–Brown approach” or the “generalizability theory approach” in past studies [8,14]. However, the accelerometer-measured PA and SB have been tested against a maximum 28 days of measurement [13]. It remains unresolved whether this is sufficient to represent individuals’ habitual physical activity over a longer period. Not knowing how long a data collection period is actually required, we would need a longer measurement period, preferably of several months, to be able to test against indisputably regular PA habits. Therefore, in this study we did not estimate the possibility of a random error in our data, but wanted to determine whether we could detect a systematic measurement effect that is possibly caused by the measurement itself. In this study, we measured PA and SB for a mean of 25 (SD 3) days. To our knowledge, there are only a few studies that have reported results of an equally long or longer wear time [13,14].

In this study, we could not detect any clear patterns indicating specific behavior changes during the four-week accelerometry, apart from the decreased sedentary time, which can largely be attributed to the declined daily wear time of the accelerometer. However, we cannot exclude the possibility that our measurement would have had an effect on the participants’ behaviors. The participants knew they were screened for a study where the target population should be sitting a lot. Although they were instructed to maintain their habitual physical activity and sedentary behaviors during the measurement and they did not know the threshold for required amount of SB to enter the intervention study, this might have had an impact on their behavior. On the other hand, one can intuitively assume that being monitored can cause a reduction in sedentary time, because being more active is generally considered a desirable behavioral trait. These two issues could have had counterbalancing effects on participants’ PA and SB. Moreover, the participants were contacted after two weeks of measurement to change the accelerometer to one with fresh batteries. This contact with the researcher may have diluted the systematic effect caused by wearing a measurement device. It is also possible that four weeks is not long enough to be fully accustomed to being monitored, which may remove the measurement effect.

The measurement effect of wearing a device is a phenomenon that is very challenging, if not impossible, to evaluate because of the lack of a control condition in study settings. However, it is not negligible, and intuitively one can assume the effect to be considerable. To our knowledge, this is the first attempt to assess systematic changes in the accelerometer-measured outcomes over time. Furthermore, when measuring human behaviors such as SB or PA, one must bear in mind that human behavior is highly adaptive, and even if rather constant in the long term, some activities are more periodical or intermittent than others. If the measurement period is too short, some critical information may remain unrecorded. Therefore, further studies evaluating sufficient duration of accelerometer data collection are warranted.

Measuring LPA is a fairly new field in PA research. The current recommendations regarding physical activity for health rely mainly on findings of the health effects of MVPA [23]. The role of LPA in health promotion is not yet fully understood. Interestingly, we observed the greatest variation between weeks 1, 2, 3 and 4 in LPA that was not diluted by calculating the proportion of wear time. There was no significant sex difference in overall LPA time or proportion, but men and women seemed to behave a little differently through the measurement weeks, whereas sex differences were observed in SB and standing time and proportions. However, in a population-based cohort, a sex difference was observed in MVPA but not in average PA intensity, indicating that women accumulated more LPA [24]. Women spending more time standing and in LPA could be explained by traditional gender roles, for example, and related household activities.

Our findings confirm that measured SB is sensitive to the daily wear time (hours/day) of the accelerometer [25]. However, the wear time decreased after the first week of measurement, but measured sedentary time did not decrease until after three weeks. Therefore, the decreases in measured sedentary time and wear time were not completely linear. However, it can be presumed that non-wear time during waking hours is most likely sedentary time. Therefore, in the future, accelerometry for a full 24 h per day would likely produce more robust and reliable measures of hourly SB and PA and their associations to health-related outcomes. In this study we did not estimate how many hours per day should be measured, but all days with data collected for 10 h or more were considered valid and included in the analyses. It can be assumed that longer daily measurement time can reduce the number of days needed for a reliable estimate of one’s PA and SB behaviors during the data collection. However, measuring longer days does not remove the possible systematic error caused by wearing the device.

Moreover, the time of year during which the measurement is taken should also be taken into account, especially in countries with considerable seasonal variation in weather [26,27]. The weather of different seasons can be a critical factor influencing personal activities. In this study, we did not monitor the same individuals at different time points and therefore individual and seasonal variation may be mixed, but we found some significant differences in sedentary time measured during winter and summer. Therefore, the time of the year when accelerometer data is collected should be carefully considered, especially if the aim is to compare groups or draw conclusions about individual PA habits and associations with health outcomes.

Interestingly, even without any identifiable, clear-cut behavior patterns during the measurement, we could not detect a significant association between measured SB and insulin resistance with accelerometer data only from the first week. However, after two weeks the association between HOMA-IR and SB proportion became significant, and after three weeks the association between HOMA-IR and LPA became significant. This strengthened association could be due to reduced data variability due to having more measured data points [20]. Measured fasting plasma insulin has great variation in the general population, and the distribution is often skewed, as was the case in this study. Moreover, the electrochemiluminescence method is sensitive to hemolysis, which can cause even more inconsistency in the results [28]. With such a sensitive measure, a more robust measure as a point of reference may be needed. On the other hand, the participants in this study did not meet the current recommendations for physical activity and they spent a major proportion of the day sedentary. This is a specific subpopulation, and linear associations may be more difficult to detect than in the general population with different activity levels. As such, our results can only be generalized to similar populations. Therefore, we recommend that SB and PA should be measured for at least three weeks if the purpose is to evaluate associations between SB and PA and insulin sensitivity or other equally delicate outcomes, especially within special populations or with a limited number of participants. Alternatively, repeated blood sampling could be considered, although it remains to be verified whether this approach would increase the consistency of the results. Altogether, there is an urgent need for studies with validated accelerometry methods to evaluate how long data collection would actually be needed to truly represent ‘habitual’ PA or SB of different study groups.

One limitation to our study is that accelerometer-based measurements are incapable of detecting the intensity or exertion in activities with low acceleration, such as carrying heavy loads. Combining accelerometry with heart rate monitoring could have additional value in such cases. Furthermore, the thigh has been adjudicated as the optimal site for the accelerometer placement when measuring SB [11]. However, the method is quite laborious and can cause discomfort and hypersensitization, especially if worn for longer periods and lengthy measurements can thus become unfeasible. Therefore, we measured PA and SB with hip-worn accelerometers for four consecutive weeks with validated analysis algorithms of PA and SB. Moreover, in the future, when reaching mass production, new techniques, e.g., wearable flexible devices, may allow multisensory techniques to be applied in field studies, and may bring new insights to PA and SB research.

## 5. Conclusions

There is considerable intraindividual variation in accelerometer-measured PA and SB, and both random and systematic errors in the measurement are possible. Therefore, if the aim is to detect the levels of participants’ habitual PA and SB, or potential changes in their behavior, the data collection should last long enough to dilute the possible bias. In this study, measured SB decreased after three weeks of measurement. Moreover, the association between SB and insulin resistance could first be detected after two weeks, and the association between LPA and insulin resistance after three weeks of accelerometry, but not based on the one-week data. Based on our findings, a measurement for at least three weeks may be needed, especially with small study samples and when the purpose is to find associations with tenuous health outcomes such as insulin resistance.

## Figures and Tables

**Figure 1 ijerph-18-04950-f001:**
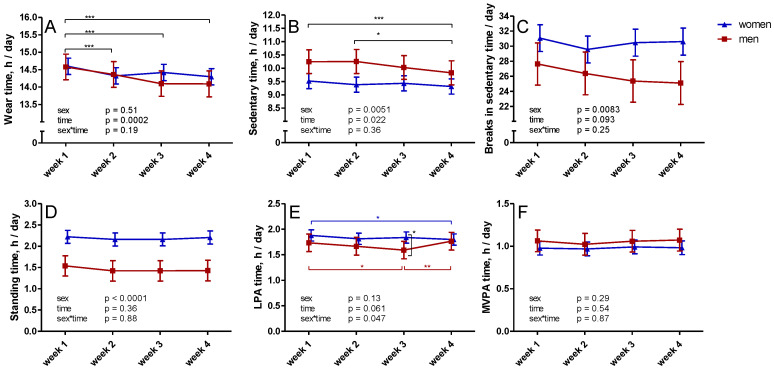
Changes in wear time of the accelerometer (**A**) and accelerometer-measured sedentary (**B**) and physical activity time (**C**–**F**) during four weeks of data collection. The mean results with 95% confidence interval of women are represented by blue triangles and men by red squares. Note: LPA—light physical activity; MVPA—moderate-to-vigorous physical activity; * *p* < 0.05; ** *p* < 0.01; *** *p* < 0.001.

**Figure 2 ijerph-18-04950-f002:**
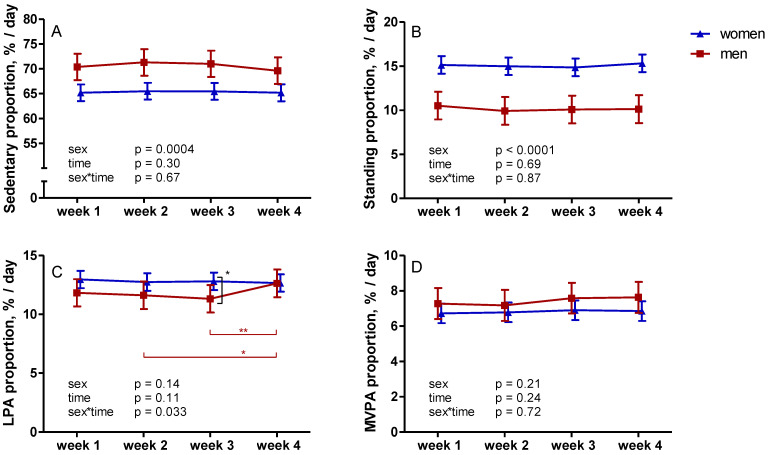
Changes in accelerometer-measured sedentary (**A**) and physical activity proportions (**B**–**D**) during four weeks of data collection. The mean results with 95% confidence interval of women are represented by blue triangles and men by red squares. Note: LPA—light physical activity; MVPA—moderate-to-vigorous physical activity; * *p* < 0.05; ** *p* < 0.01.

**Figure 3 ijerph-18-04950-f003:**
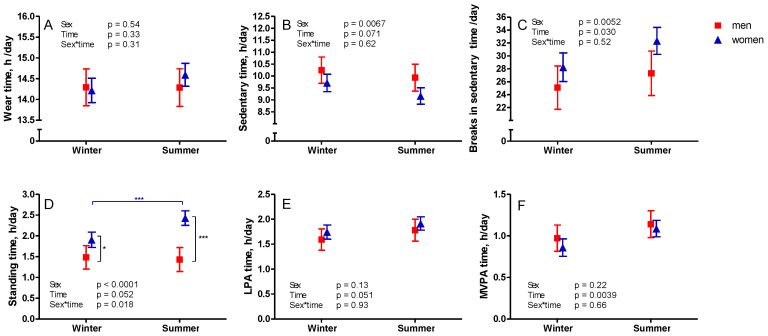
Differences in wear time of the accelerometer (**A**) and accelerometer-measured sedentary (**B**) and physical activity time (**C**–**F**) during winter (November–March) and summer (April–October). The mean results with 95% confidence interval of women are represented by blue triangles and men by red squares. Note: LPA—light physical activity; MVPA—moderate-to-vigorous physical activity; * *p* < 0.05; *** *p* < 0.001.

**Figure 4 ijerph-18-04950-f004:**
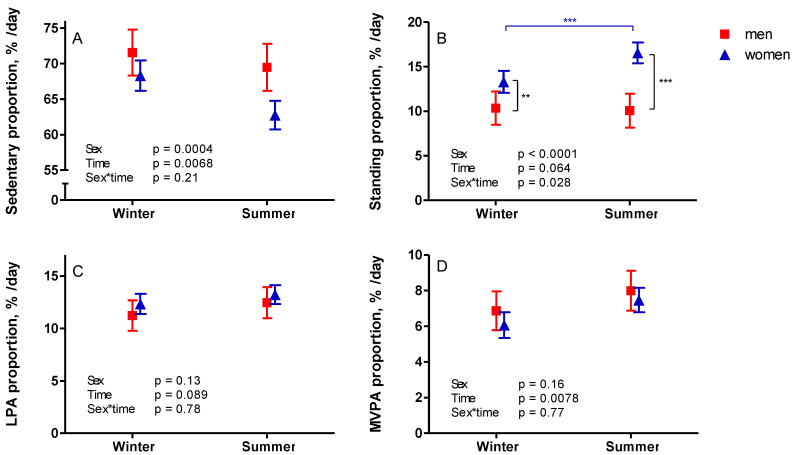
Differences in accelerometer-measured sedentary (**A**) and physical activity proportions (**B**–**D**) during winter (November–March) and summer (April–October). The mean results with 95% confidence interval of women are represented by blue triangles and men by red squares.Note: LPA—light physical activity; MVPA—moderate-to-vigorous physical activity; ** *p* < 0.01; *** *p* < 0.001.

**Table 1 ijerph-18-04950-t001:** Basic characteristics of the study participants. Unless otherwise stated the results are presented as mean (SD).

n, (% men)	143 (29)
Median age, years (IQR)	59 (9)
Body mass index, kg/m^2^	31.7 (4.0)
Waist circumference, cm	109.5 (11.4)
Fasting plasma glucose, mmol/l	5.8 (0.8)
Median fasting plasma insulin, mU/l (IQR)	11 (8)
Median HOMA-IR (IQR)	2.7 (2.2)
Antihypertensive medication, n (%)	56 (39)
Cholesterol lowering medication, n (%)	19 (13)

**Table 2 ijerph-18-04950-t002:** Cumulative means of accelerometer measures (h/day) during weeks 1–4 and associations with homeostatic model assessment for insulin resistance (HOMA-IR) analyzed with linear models with age, sex, and body mass index (BMI) included in the model. Sex and BMI were significant in all of the models, with BMI being the strongest predictor of HOMA-IR.

	Duration, h/day (SD)	Measurement Duration, days (SD)	Estimate, B	*p*
Sedentary time, h/day				
1 week	9.73 (1.47)	6.7 (0.7)	0.01	0.45
2 weeks	9.68 (1.39)	12.8 (1.8)	0.01	0.34
3 weeks	9.65 (1.36)	19.1 (2.5)	0.01	0.32
4 weeks	9.61 (1.32)	24.9 (3.5)	0.02	0.24
Standing time, h/day				
1 week	2.02 (0.86)	6.7 (0.7)	−0.03	0.17
2 weeks	1.98 (0.79)	12.8 (1.8)	−0.03	0.21
3 weeks	1.97 (0.76)	19.1 (2.5)	−0.04	0.15
4 weeks	1.97 (0.76)	24.9 (3.5)	−0.03	0.23
LPA time, h/day				
1 week	1.84 (0.59)	6.7 (0.7)	−0.05	0.13
2 weeks	1.80 (0.54)	12.8 (1.8)	−0.06	0.073
3 weeks	1.79 (0.52)	19.1 (2.5)	−0.07	0.041
4 weeks	1.79 (0.50)	24.9 (3.5)	−0.07	0.042
MVPA time, h/day				
1 week	1.01 (0.39)	6.7 (0.7)	−0.12	0.017
2 weeks	0.99 (0.38)	12.8 (1.8)	−0.12	0.017
3 weeks	1.00 (0.38)	19.1 (2.5)	−0.13	0.010
4 weeks	1.00 (0.38)	24.9 (3.5)	−0.14	0.0065

**Table 3 ijerph-18-04950-t003:** Cumulative means of accelerometer measures (% of wear time) during weeks 1–4 and associations with homeostatic model assessment for insulin resistance (HOMA-IR) analyzed with linear models with age, sex, and body mass index (BMI) included in the model. Sex and BMI were significant in all of the models, with BMI being the strongest predictor of HOMA-IR.

	Mean %/day (SD)	Measurement Duration, days (SD)	Estimate, B	*p*
Sedentary proportion, %/day				
1 week	66.6 (8.9)	6.7 (0.7)	0.42	0.055
2 weeks	67.0 (8.6)	12.8 (1.8)	0.46	0.044
3 weeks	67.0 (8.3)	19.1 (2.5)	0.52	0.026
4 weeks	66.8 (8.1)	24.9 (3.5)	0.55	0.023
Standing proportion, %/day				
1 week	13.8 (5.6)	6.7 (0.7)	−0.40	0.27
2 weeks	13.6 (5.3)	12.8 (1.8)	−0.40	0.30
3 weeks	13.6 (5.1)	19.1 (2.5)	−0.48	0.24
4 weeks	13.6 (5.0)	24.9 (3.5)	−0.43	0.30
LPA proportion, %/day				
1 week	12.7 (3.9)	6.7 (0.7)	−0.64	0.18
2 weeks	12.5 (3.6)	12.8 (1.8)	−0.85	0.095
3 weeks	12.5 (3.5)	19.1 (2.5)	−1.00	0.060
4 weeks	12.5 (3.4)	24.9 (3.5)	−1.04	0.054
MVPA proportion, %/day				
1 week	6.9 (2.6)	6.7 (0.7)	−1.49	0.041
2 weeks	6.9 (2.6)	12.8 (1.8)	−1.57	0.033
3 weeks	7.0 (2.6)	19.1 (2.5)	−1.68	0.023
4 weeks	7.0 (2.6)	24.9 (3.5)	−1.87	0.012

## Data Availability

The datasets generated during the current study are available from the corresponding author on reasonable request.

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
