# Peer review of "Influence of the Duration and Timing of Data Collection on Accelerometer-Measured Physical Activity, Sedentary Time and Associated Insulin Resistance"

_ijerph, 2021, doi:10.3390/ijerph18094950_

Round 1
Reviewer 1 Report
The paper discussed the effective duration and timing of data collection of accelerometry in getting the associations between HOMA-IR and SB/PA. The research is helpful, but the results are, somehow, not strong enough to show the intend conclusions. Researches on human activities are hard works, and many deviations may happen.
Major comment
- All the participants sit a major proportion of their day, but the occupations may influence activities when they are sitting down, causing different body responses.
- More information about the worn device is needed to prove that the activities are not influenced by the additional device, e.g. size, weight, comfort level, etc.
3.The weather of different seasons can be a big influence factor in personal activities. In my opinion, it is better to show which data are collected on which season. Additional discussions about it are also needed.
- The begin and end time for data collection in everyday may also influence the results. A simple description about it is needed.
Minor comments
- Clerical error in Lines 217-221.
2. New techniques for monitoring personal activities, e.g. wearable flexible devices, can be a great revolution (Many related papers have been published in MDPI journals, e.g. Sensors). A discussion or prediction can be helpful .
Reviewer 2 Report
The paper presents a study that evaluates possible changes in the measured outcomes and associations with insulin resistance during four weeks of accelerometry. Overall, the paper is well written and an appropriate methodology. However, the introduction and discussion need to be more persuasive, as well as the practical implications of the present study. I include some comments below related to this summary for consideration.
- In relation to the contribution of the study to the literature, I did not get a sense from the article that the findings revealed anything other than what we already know. In this regard, the authors need to justify in better detail the purpose of the present study. What is the gap, which you need to be addressed?
- The introduction of the paper was very descriptive, it did not situate the current study in literature or highlight what the gap in the literature is that this study is trying to address. At least, the authors should situate better the main purposes of this study;
- Methodology
- The methods section lacks – General concerns
- the recruitment date range (month and year);
- 2a description of any inclusion/exclusion criteria that were applied to participant recruitment;
- a table of relevant demographic details;
- a statement as to whether your sample can be considered representative of a larger population,
- 5a description of how participants were recruited, and descriptions of where participants were recruited and where the research took place.
- Specific concerns:
- What was the criterion for organizing the subjects into those three groups?
- How did you determine the power of sample size?
- What is the practice level?
- How many years of practice do the subjects have?
- Please include the power of sample size;
- Please include the partial eta-square;
- Did you perform any analysis regarding outliers?
- Overall the discussion is very descriptive and any statements about the contribution and conclusions of the study are not new, in their current form.
- Lack of theoretical and empirical connection between the constructs analyzed, as well as, between present findings and previous studies. The practical implications need to be further explored,
- The discussion is very descriptive and any statements about the contribution and conclusions of the study are not new. At least this moment. Please clarified better and justified your choices.
- Overall, the paper has conditions for being accepted in IJERPH, however, the authors should clarify the points above.
Reviewer 3 Report
In general, this manuscript is easy to follow. I just would like to suggest to strengthen the link among insulin resistance, the use of different moments in the year, with the main aim of the study.
Abstract
Line 19 – only men were characterized?
Introduction
This section is easy to follow and show the study rational until line 83, where the Model Assessment for Insulin Resistance is indicated but any description is made.
Methods
Line 134 – which was the overall percentage included in the study? It was representative?
Why measure during winter and during summer? What are the possible influences?
Please give more information about statistics: do you use slope? Intercept, variables included
It was made any questionnaire/ anamneses?
Results
Line 167-173 – Unless you applied a questionnaire, it seems that this information could be included in the method session.
Graphics really helped to read results, they are very clean and expressive.
Discussion
Line 254 – how long?
Round 2
Reviewer 1 Report
The revision is great, the paper can be accepted now.
Reviewer 2 Report
No more comments.